# Studying the Influences of Bus Stop Type and Specifications on Bicycle Flow and Capacity for Better Bicycle Efficiency

**Xingchen Yan [1], Jun Chen [2,*], Xiaofei Ye [3], Tao Wang [4], Zhen Yang [1] and Hua Bai [5]**

[1] College of Automobile and Traffic Engineering, Nanjing Forestry University, Longpan Road 159#, Nanjing 210037, China; xingchenyan.acad@gmail.com (X.Y.); zyang_2016@163.com (Z.Y.)

[2] School of Transportation, Southeast University, Dongnandaxue Road 2#, Jiangning Development Zone, Nanjing 211189, China

[3] School of Maritime and Transportation, Ningbo University, Fenghua Road 818#, Ningbo 315211, China; yexiaofei@nbu.edu.cn

[4] School of Architecture and Transportation, Guilin University of Electronic Technology, Jinji Road 1#, Guilin 541004, China; wangtao_seu@163.com

[5] China Design Group Co., Ltd., Ziyun Road 9#, Nanjing 210014, China; bh_birch@163.com

\* Correspondence: chenjun@seu.edu.cn; Tel.: +86-139-1394-5222

**Abstract:** This study aimed to explore the effects of type and specifications of bus stop on bicycle speed and cycle track capacity. This paper investigates the traffic flow operations of tracks at basic sections, curbside stops, and bus bays by video recording. T-test and comparative study were used to analyze the influences of stop types on bicycle speed and capacity of track. The relationships between stop specifications and speed and capacity of track are analyzed with correlation analysis. The main results are as follows: (1) Without passengers crossing, bus bays have significant impact on bicycle speed, while it is not for curbside stops; (2) except platform length, there are strong negative relationships between bicycle speed and density of platform access, total width of platform accesses (TWPA), total width of platform accesses-to-platform length ratio (TWPA-to-PL ratio), total width of platform accesses-to-track width ratio (TWPA-to-TW ratio); (3) curbside stop and bus bay reduce track capacities by 32% and 13.5% on average, respectively; and (4) in contrast to bus bays, curbside stops have more significant impact on capacity of track, which also presents in the influence of the setting parameters of stops. Based the results above, some suggestions on stop specifications are finally proposed.

**Keywords:** bus stop; bicycle flow; impact of stop layout; correlation analysis; stop specifications

## 1. Introduction

In order to improve the sustainability and livability of urban environment, cycling and public transport, due to their low energy consumption and low pollution, have been strongly encouraged in metropolitan areas worldwide [1,2]. Various kinds of bicycle facilities and public transport infrastructures are being planned and constructed in Europe, the United States and China's cities. Bicycle and bus travel have been rapidly improved and developed [3–6].

In developing countries, such as China and India, bicycling and public transport have long been important modes of commuting because of low-income levels. After a large number of bicycle facilities and bus stops were built on the streets, some problems of mutual interference between the two modes present. In China, bicycle lanes are installed on both sides of the streets, bus platforms are set on

the sidewalks, and buses pull over to the curbside and occupy bicycle lanes to dwell. In this case, the mixed traffic flow leads to traffic conflicts, speed drop and even traffic jams.

In order to solve the problems due to the mixtures of buses and cycles mentioned above, bike lane or cycle track is physically re-routed outside the main carriage way, around the bus stop. It is called bus stop bypass (BSB) where cyclists pass between a bus stop platform and the main pedestrian footway via a cycle track. There are two common types of BSB layout (see Figure 1a,c). The first type of layout is continuously straight as shown in Figure 1a,b. The amenities are aligned with the back of platform, dividing the back borders into some accesses. The second has a 1-way cycle track that kinks around the back of a bus stop platform where the amenities and accesses are located as indicated in Figure 1c,d.

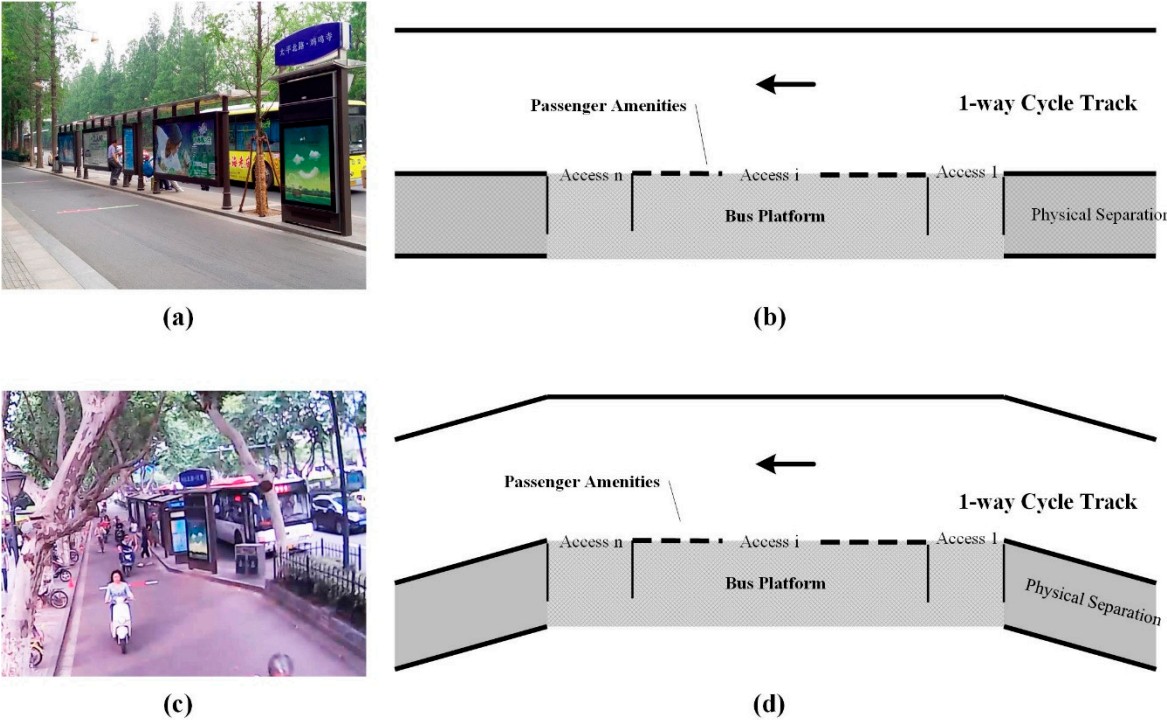

**Figure 1.** (**a**) Field picture (Jimingsi bus stop) for curbside bus stop with 1-way cycle track; (**b**) layout for curbside bus stop with 1-way cycle track; (**c**) field picture (Gulou bus stop) for bus bay with 1-way cycle track; (**d**) layout for bus bay with 1-way cycle track.

Previously, numerous studies evaluating the traffic flow characteristics in the vicinity of bus stops have been conducted. Koshy et al. compared the influence of the curbside stops and bus bays on the operations of motorized vehicles [7]. Zhao et al., evaluated the traffic interactions between the motorized and non-motorized vehicles near a bus stop station [8]. Sun and Elefteriadou analyzed the characteristics of vehicle lane-changing behaviors near the bus stops [9]. In Zhao et al.'s paper, the interactions between the buses and the bicycles at different types of bus stops were evaluated [10]. Zhang et al. investigated the influences of four types of bus stops on bicycles, vehicles, and buses [11]. Wang et al. evaluate the effects of dwelling buses on the traffic operations of nonmotor vehicles at stops in China [12]. Luo et al. used gap theory and queuing theory to model the effect of bus stops on capacity of curb lane [13]. Pan et al. explored the potential factors affecting the length of bus stop influence zone using multi-source data [14]. In respect to bus stop layout optimization, Liu et al. conducted a simulation experiment to find the ideal bus stop layout facilitating arterial coordinated control [15]. Tirachini tried to determine the numbers of bus stop along urban routes by estimating the probability of stopping in low demand markets, and by analyzing the interplay between bus stop size, bus running speed, spacing and congestion in high demand markets [16]. In additional to the scholars, many local authorities and organizations proposed guidelines or standards on bus stop design [17–22].

By reviewing the recent literatures, we found that (1) most scholars conducted their studies under the conditions of mixed operation of motor vehicles and bicycles in street; (2) meanwhile, they pay more attention to the impact of bus stops on private cars and capacity of carriageway; (3) On bus stop specifications, researchers and guidelines focused on locations and spacing from a perspective of a bus line or a street. In the trend of encouraging more dedicated cycling facilities and bus stops, research strength need to be diverted to the influence of bus stops on cycle tracks or protected bicycle lanes.

This study mainly analyzes the impact of bus stop type and geometric design on bicycle flow running on the BSB and capacity of BSB without passengers entering and existing. Based on the results, suggestions on the basic problems of bus stop design are proposed. The study investigated the two types of BSB in Figure 1 and bicycle flow consisted of electric bikes (EB) and conventional bikes (CB). The remaining sections of the paper are organized as follows. The section "Methodology" describes the proposed methods to collect field data, compare the changes in cycle speeds, and analyze the relationships between stop specifications and bicycle speed, track capacity. The following two sections show the results. The findings and conclusions are presented in the last section.

## 2. Methodology

The study mainly includes three phases as shown in Figure 2: **(1) Field investigation** consists of field measuring and video recording. The former gets the type, geometric dimensions and other information about bus stop specifications, while the latter records bicycle flow running at the collected bus stops. **(2) Data processing** means basic data, like bike speed and flow rate, is extracted from the record videos by manual counting and bicycle volume, density and the parameters of stop specification are obtained through parameter calculation. The two phases above are introduced in Section 2.1 using illustrative charts and equations. **(3) Impact analysis** performs parametric effect analysis on bike speed and track capacity with T-test and correlation analysis which are described in Sections 2.2 and 2.3.

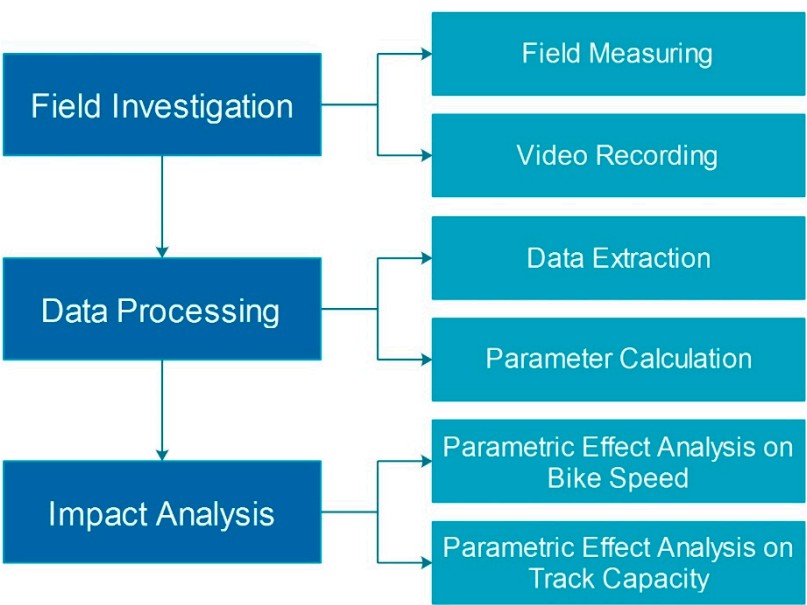

**Figure 2.** Flow chart of the study.

### 2.1. Data Collection

#### 2.1.1. Field Survey

To study the impact of bus stop on bicycle flow, it needs compare the bicycle operations at stops with those at basic segments. Therefore, the geometric dimensions of cycle track and bus stop are

included in our field survey, and then the bicycle operation data at the facilities mentioned above is necessary.

We investigated bicycle flow operations using a dual-camera recording method during the workdays in Nanjing. Field observations were performed in commuting intervals, from 17:00 to 9:00 for the morning peak and from 17:00 to 19:00 for the evening peak. The following are the selection criteria for the survey sites:

(1)　differences in bicycle track width;
(2)　paved level terrain, good sight;
(3)　far from intersections, block accesses and bus stations;
(4)　differences in type of bus stop;
(5)　the compared basic segments located far from the bus stops;
(6)　suitable space for installing cameras;

According to our preliminary observation, we chose 50 m as the length of a survey segment. For BSB, 25 m downstream and downstream of the central line of the platform are selected.

The instruments used in our field work included a tape measure of 50 m, two wide-angle cameras, two tripods (maximum height of 4 m), 12 red traffic cones, and six marking tapes (dimensions: 30 cm × 150 cm and 120 cm of the center part colored in red).

The setting details for the observation are described in Figure 3a–d.

### 2.1.2. Data Extraction

The research takes 10 s as time unit to extract bicycle operation data from video, including bicycle type, speed, lateral position, volume and density. All data fragments are divided into two categories: with bus dwelling and passengers crossing, and without. Because this paper studies the influence of platform specifications on bicycle speed and track capacity, only the second kind of data is used.

When a bicycle travels through the observation area, the observer needs to record the following information: bicycle type, the time when the bicycle reaches each marking tape $t_i$ ($i = 1, 2, \ldots, 6$), lateral positions of the bicycle at each tape $lp_i$ ($i = 1, 2, \ldots, 6$). Lateral positions are identified according to the description in Figure 3c. The calculating equations of parameters analyzed later are indicated as follows.

Speed:

$$v_n^{ij} = \frac{10}{t_j - t_i},\tag{1}$$

where $v_n^{ij}$ = speed of bicycle n between tape $i$ and tape $j$; and $t_i$, $t_j$ = moments that bicycle $n$ reached tape $i$ and tape $j$.

Volume: the number of bicycles passing the $j$th tape was counted at an interval of 10 s and the mean volume was computed by the following equation.

$$\overline{q}_j = \frac{q_j}{w} \cdot \frac{3600}{10},\tag{2}$$

Density: The number of vehicles between the marking tape $i$ and $j$ was recorded every 2 s, and the average density was obtained every 10 s using Equation (3).

$$\overline{d}_{ij} = \frac{1}{5} \cdot \frac{N_{ij}^1 + N_{ij}^2 + \cdots + N_{ij}^5}{10 * w} \cdot 1000,\tag{3}$$

Capacity: the bicycle flow-density relationship was fitted by Greenshield's model and track capacity was obtained at the maximum of the fitting curve. Greenshield's flow-density model is shown as follows.

$$q = A \cdot k - B \cdot k^2,\tag{4}$$

where:

$q$ = flow (bics/h/m)
$A$, $B$ = constants
$k$ = density (bics/km/m).

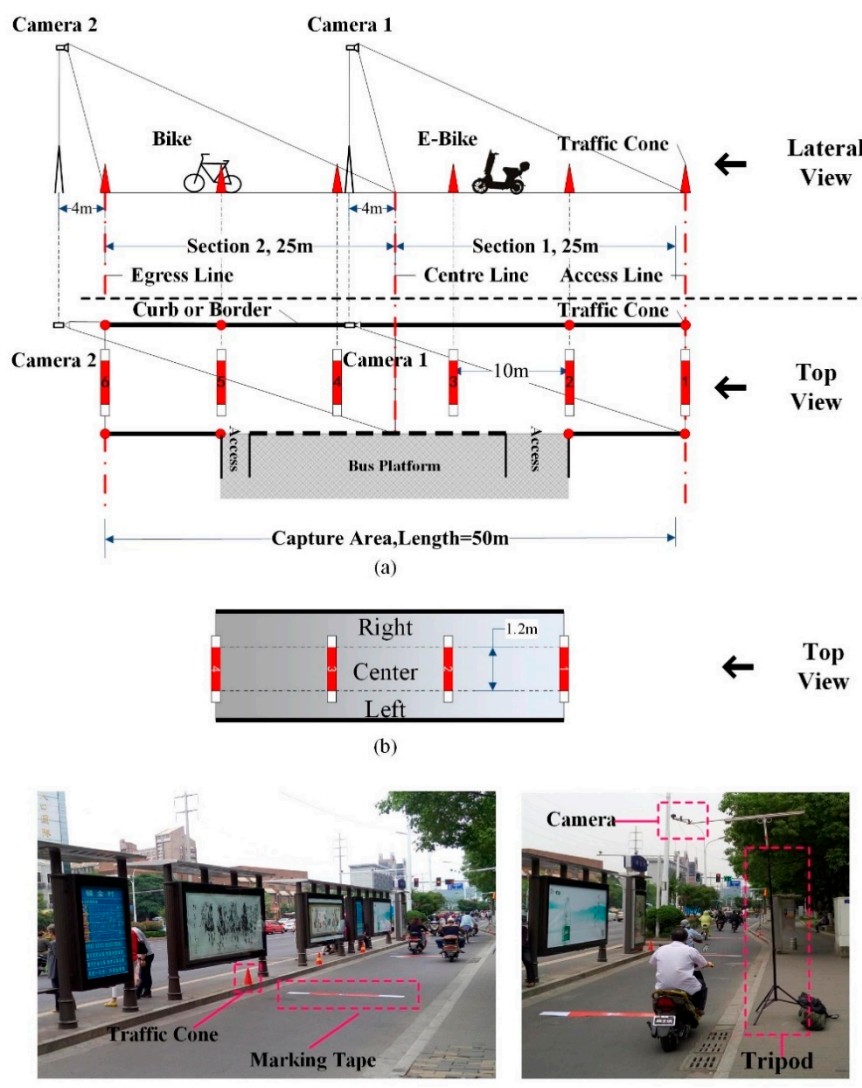

**Figure 3.** (**a**) Setting details for field observation; (**b**) track surface markings; (**c**) installation of traffic cones and marking tapes; and (**d**) setup of the camera and tripod in field.

### 2.1.3. Results and Descriptive Analysis

In addition to the basic parameters of a BSB, we defined the following new parameters to study the combined effects of two basic parameters.

Density of Accesses (DA, unit: counts/m) is defined as number of access ($n_a$) divided by platform length (PL):

$$DA = \frac{n_a}{PL}, \tag{5}$$

Total Width of Platform Accesses (*TWPA*, unit: m) is the sum of all access widths of a platform:

$$TWPA = w_{a1} + w_{a2} + \cdots + w_{an}, \tag{6}$$

where $w_{an}$ = width of access n.

Total Width of Platform Accesses-to-Platform Length Ratio (*TWPA* to *PL* ratio) is defined as the proportion that total width of platform accesses covers platform length.

$$TWPA \ to \ PL \ ratio = \frac{TWPA}{PL}, \tag{7}$$

Total Width of Platform Accesses-to-Track Width Ratio (*TWPA* to *TW* ratio) is defined as total width of platform accesses (*TWPA*) divided by track width (*w*):

$$TWPA \ to \ TW \ ratio = \frac{TWPA}{w}, \tag{8}$$

This study investigated six BSBs, including 3 curbside stops and 3 bus bays. Table 1 shows the specification information of six BSBs. Two basic segments of cycle track were surveyed. One located at Nanjing blood center and the other at Baima Park, with 3 m and 4 m of track, respectively.

**Table 1.** Specifications of six bus stops.

| Number | Stop | Stop Type | TW (m) | PL (m) | * $N_a$ | Access Position | DA | TWPA | TWPA to PL Ratio | TWPA to TW Ratio |
|---|---|---|---|---|---|---|---|---|---|---|
| 1 | Gangzicun | curbside | 3 | 40 | 4 | two ends and center | 0.10 | 14.9 | 0.37 | 4.97 |
| 2 | Zhongyangmennan | curbside | 3.5 | 59.8 | 7 | two ends | 0.03 | 19.1 | 0.32 | 5.46 |
| 3 | Jimingsi | curbside | 4 | 30 | 4 | two ends and center | 0.13 | 15.8 | 0.53 | 3.95 |
| 4 | Suojincun | bus bay | 3 | 51 | 5 | two ends and center | 0.10 | 15.9 | 0.31 | 5.30 |
| 5 | Gulou | bus bay | 3.5 | 37.3 | 2 | irregular | 0.19 | 19 | 0.51 | 5.14 |
| 6 | Zhongyangmenbei | bus bay | 4 | 54.7 | 5 | two ends and center | 0.09 | 15.7 | 0.29 | 3.93 |

* $N_a$ = number of accesses.

Table 2 presents a summary of the analytical results of the bicycle vehicle speeds at different locations for different types of bus stops. More than 200 bicycle samples were selected randomly for each bus stop to analyze the speed characteristics, including maximum value, minimum value, mean value, median value, and standard variation. The arithmetic means of bicycle speeds for BSBs at curbside stops were 6.14 ± 1.24, 5.29 ± 1.05, and 5.82 ± 1.65 m/s; and 5.22 ± 1.07, 4.39 ± 0.99, and 5.17 ± 1.19 m/s for BSBs at bus bays, respectively. The capacities for BSBs at curbside stops were 1984, 2523, and 1325 bics/m/h; and 2114, 1688, and 2055 bics/m/h for BSBs at bus bays, respectively.

**Table 2.** Bicycle speeds and track capacities at six bus stops.

| Number | Bicycle Type | Sample Size | Minimum (m/s) | Maximum (m/s) | Mean (m/s) | Median (m/s) | SD (m/s) | Capacity (bics/m/h) |
|---|---|---|---|---|---|---|---|---|
| 1 | EB | 598 | 3.59 | 10.95 | 6.33 | 6.28 | 1.13 | |
| | CB | 81 | 2.71 | 7.46 | 4.7 | 4.75 | 1.10 | 1984 |
| | overall | 679 | 2.71 | 10.95 | 6.14 | 6.15 | 1.24 | |
| 2 | EB | 591 | 1.98 | 8.49 | 5.58 | 4.50 | 1.00 | |
| | CB | 158 | 1.84 | 5.46 | 4.19 | 3.35 | 0.63 | 2523 |
| | overall | 749 | 1.84 | 8.49 | 5.29 | 4.27 | 1.05 | |
| 3 | EB | 317 | 2.82 | 10.53 | 6.12 | 6.14 | 1.52 | |
| | CB | 67 | 2.55 | 9.67 | 4.39 | 4.17 | 1.38 | 1324 |
| | overall | 384 | 2.55 | 10.53 | 5.82 | 5.78 | 1.65 | |
| 4 | EB | 667 | 2.65 | 9.42 | 5.33 | 5.32 | 1.02 | |
| | CB | 73 | 2.81 | 6.51 | 4.17 | 4.22 | 0.87 | 2114 |
| | overall | 740 | 2.65 | 9.42 | 5.22 | 5.24 | 1.07 | |
| 5 | EB | 1007 | 3.21 | 11.88 | 4.58 | 5.18 | 0.99 | |
| | CB | 194 | 2.41 | 7.34 | 3.43 | 4.81 | 0.89 | 1688 |
| | overall | 1201 | 2.41 | 11.88 | 4.39 | 5.12 | 0.99 | |
| 6 | EB | 858 | 2.80 | 9.65 | 5.24 | 5.90 | 1.07 | |
| | CB | 154 | 2.36 | 7.91 | 4.81 | 4.38 | 0.95 | 2055 |
| | overall | 1012 | 2.36 | 9.65 | 5.17 | 5.68 | 1.19 | |

## 2.2. Testing Speed Differences

The Student's t-test is often used to test the significance of the differences between two means from two different samples. Let $\mu_1$ and $\mu_2$; and $S_1$ and $S_2$ be the mean bicycle speeds and variance of bicycle speeds at two different sites, respectively. The null hypothesis states that the two means are equal:

$$H_0 : \mu_1 = \mu_2, \tag{9}$$

versus

$$H_1 : \mu_1 \neq \mu_2, \tag{10}$$

can be rejected if

$$Z^* = \frac{(\overline{x_1} - \overline{x_2}) - (\overline{\mu_1} - \overline{\mu_2})}{\sqrt{\frac{s_1^2}{n_1} + \frac{s_2^2}{n_2}}}, \tag{11}$$

where $n_1$ and $n_2$ = sample sizes for two different sites; $\alpha(\alpha = 0.05)$ = level of significance, and $Z_{\alpha/2} = 100(1 - \alpha/2)\%$ percentile of standard normal distribution.

## 2.3. Analyzing the Relationships Between Stop Specifications and Bicycle Speed, Track Capacity

When investigating a relationship between two variables, the first step is to show the data values graphically on a scatter diagram. On a scatter diagram, the closer the points lie to a straight line, the stronger the linear relationship between two variables. To quantify the strength of the relationship, we can calculate the correlation coefficient. In algebraic notation, if we have two variables $x$ and $y$, and the data take the form of n pairs (i.e., $(x_1, y_1)$, $(x_2, y_2)$, $(x_3, y_3)$ ... $(x_n, y_n)$), then the correlation coefficient is given by the following equation:

$$r = \frac{\sum\limits_{i=1}^{n} (x_i - \overline{x})(y_i - \overline{y})}{\sqrt{\sum\limits_{i=1}^{n} (x_i - \overline{x})^2 \sum\limits_{i=1}^{n} (y_i - \overline{y})^2}}, \tag{12}$$

where $\overline{x}$ is the mean of the x values, and $\overline{y}$ is the mean of the $y$ values. The criteria for strength of association between $x$ and $y$ are shown in Table 3.

**Table 3.** Criteria for Strength of Association.

| Strength of Association | Coefficient, $r$ | |
|:---:|:---:|:---:|
| | **Positive** | **Negative** |
| Small | 0.1 to 0.3 | −0.1 to −0.3 |
| Medium | 0.3 to 0.5 | −0.3 to −0.5 |
| Large | 0.5 to 1.0 | −0.5 to −1.0 |

## 3. Impact of Bus Stop Type and Specifications on Bicycle Speed

### 3.1. Stop Type

Table 4 shows the results of bicycle speed comparison among basic segment, curbside stop, and bus bay with cycle tracks of 3 m and 4 m wide.

It reveals the differences in speed between tracks at basic segments and curbside stops are not statistically significant (0.742 and 0.913). However, the differences are significant between tracks at basic segments and bus bays, as well as curbside stops and bus bays. It mainly results from the track alignment upstream of bus bay. The curved curb has strong impact on bicycle speed.

Due to the significant impact bus bay imposes on bicycle speed, the following will further analyze the relationships between the specifications of bus platform and speed.

**Table 4.** *t*-Tests Results of the Differences in Speeds at Different Cycle Track Widths.

| Track Width | Basic Segment vs. Curbside Stop | Basic Segment vs. Bus Bay | Curbside Stop vs. Bus Bay |
|---|---|---|---|
| 3 m | 0.742 (No) | 0.000 (Yes) | 0.001 (Yes) |
| 4 m | 0.913 (No) | 0.001 (Yes) | 0.003 (Yes) |

### 3.2. Number, Location, and Density of Bus Platform Accesses

In general, fewer accesses and more regular distribution of them are helpful to cyclists to observe the situation around the platform. In this case, the potential impact of accesses on the speed of bicycles is also smaller. In order to examine the combined impact of the number and location of platform accesses, the relationship between bike speed and access density was analyzed. Figure 4a confirms the perception just put forward. There is a strong negative correlation between the speed of EB, CB and the density of platform access. The coefficient R is 0.985 and 0.913 for EB and CB, respectively.

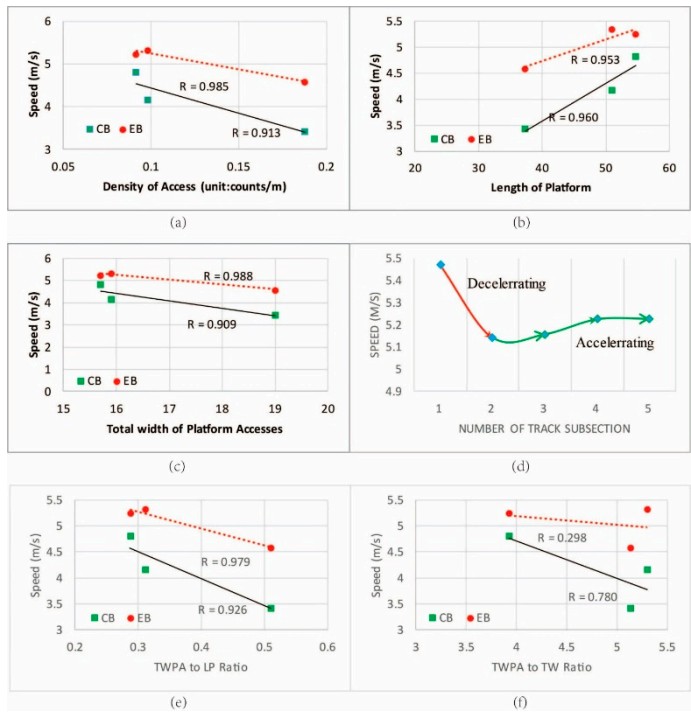

**Figure 4.** (**a**) Relationships between Bicycle Speed and Density of Platform Access; (**b**) Relationships between Bicycle Speed and Platform Length; (**c**) Trend of Mean of bike Speed travelling through bypass of Gulou Stop; (**d**) Relationships between Bicycle Speed and Total width of Platform Accesses; (**e**) Relationships between Bicycle Speed and total width of platform accesses-to platform length (TWPA-to-PL) Ratio; (**f**) Relationship between Bicycle Speed and TWPA-to-total width (TW) Ratio.

### 3.3. Platform Length and TWPA

Figure 4b shows that there is a strong positive correlation between the average speed of bicycle and platform length. This can be fully explained by analyzing the average speed change of bicycle driving through five subsections within the scope of investigation. As shown in Figure 4d, bicycle speed has a process of falling first and then rising. Due to the curved alignment upstream of bus platform, speed reduces fast and then recovers slowly in the following four subsections. If the following part is long enough, speed will return to the level that a bicycle drive in the observed area. Therefore,



when the platform length is longer, the higher the speed of bicycle driving out of the platform area is, which contribute to shift the mean speed of the process.

Platform accesses are the interface for bus passengers to cross cycle tracks. Wider access will make riders feel greater potential lateral interference when passing, so they lower the speed of riding. The scatter distribution and fitting results in Figure 4c fully illustrate this point.

### 3.4. TWPA to PL Ratio and TWPA to TW Ratio

TWPA to PL ratio is an effective proportion of platform length to disturb a bicycle flow. According to the analysis of the influence of TWPA on bicycle flow, a larger proportion will have greater impact on the speeds of cyclists. Figure 4e shows that there is a strong negative correlation between the average speed of EB, CB, and TWPA to PL ratio (R is 0.988 and 0.909, respectively).

Figure 4f reveals a large negative relationship between TWPA to TW ratio and EB speed, while it is weak for CB. It is necessary to collect more data to confirm the TWPA to TW ratio's influence on bicycle speed.

## 4. Impact of Bus Stop Type and Specifications on Track Capacity

### 4.1. Stop Type

In order to examine the influence of stop type on track capacity, we compared the flow-density relationships of 3 m and 4 m of cycle tracks at basic segments, curbside stops, and bus bays, as shown in Figure 5a,b.

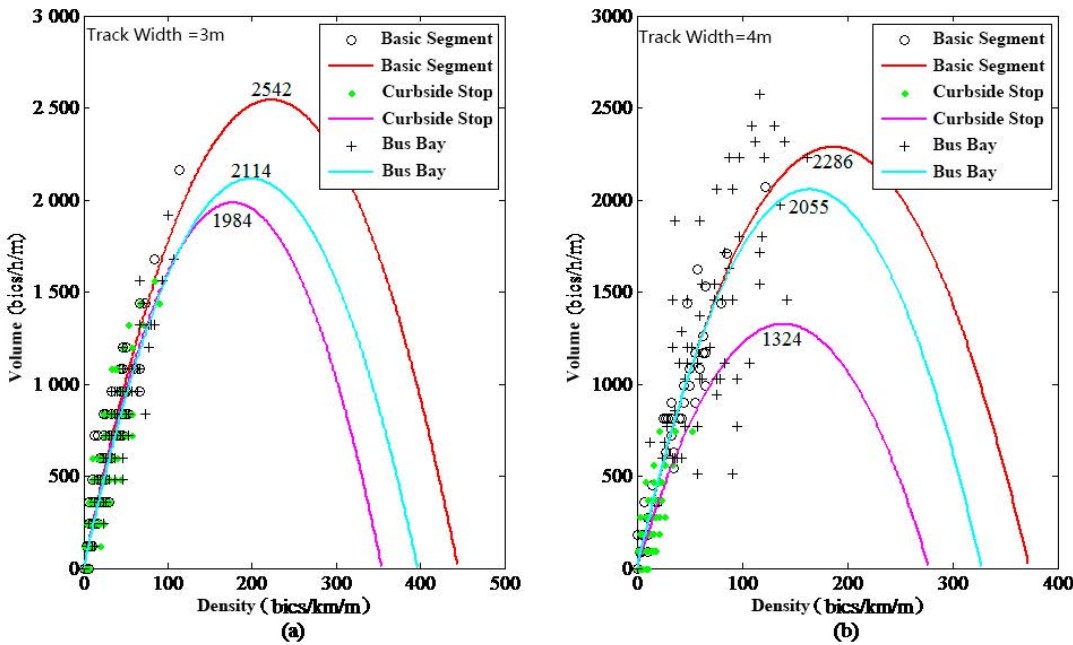

**Figure 5.** Fitting Results of Capacities Track of (**a**) 3 m and (**b**) 4 m wide at Basic Segment, Curbside Stop, and Bus Bay.

From the fitting capacity, it can be seen that curbside bus stop has a greater impact on capacity of track than bus bay does. The two types of stops reduce track capacities by 32% and 13.5% on average, respectively. A problem turns up when we review the impact of stop type on bicycle speed. In contrast to curbside stop, bus bays have significantly influenced bicycle speed. However, the findings are opposite in capacity comparison.

To answer the problem above and represent the effects of bus stops on bicycle operations, lateral positions of bikes running on bypasses were compared among basic segments, curbside bus stops, and bus bays.

(1)　From Figure 6a,b, on basic segments of cycle tracks, the occupancy rates of left, center, and right part are about 23%, 52%, and 25%. Although there are little fluctuations in the process of moving forward, these rates keep basically stable.

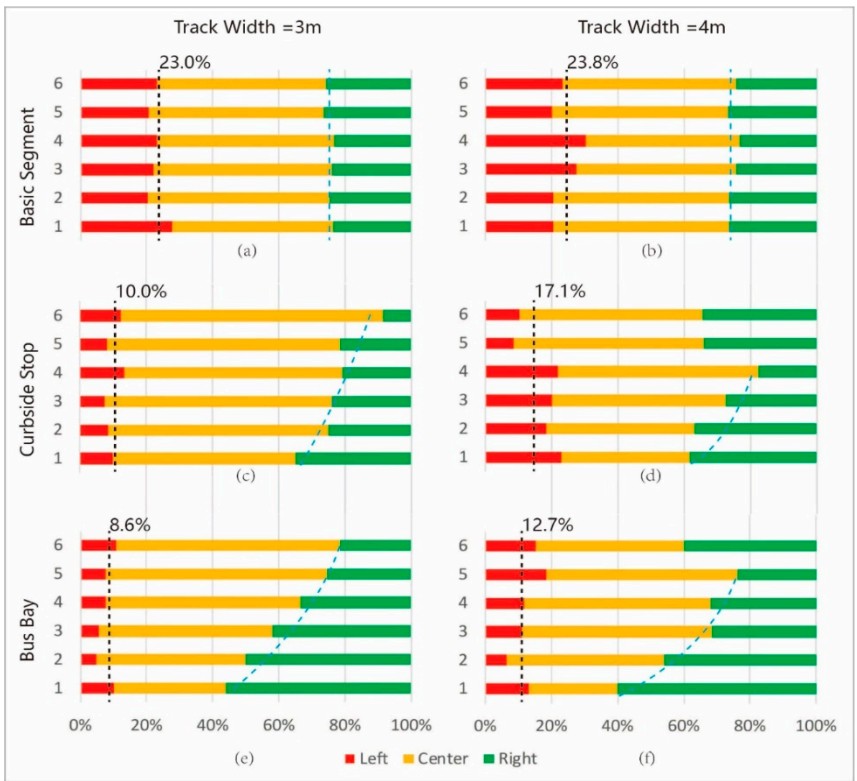

**Figure 6.** Differences in bike lateral distribution on (**a**) 3 m and (**b**) 4 m of tracks at basic segments, (**c**) 3 m and (**d**) 4 m of bypasses at curbside stops, and (**e**) 3 m and (**f**) 4 m of bypasses at bus bays.

(2)　Figure 6c–f show that the occupancies of the left side of cycle tracks decrease to less than 20%, which is also the most significant difference between the two types of bus stops and basic segments. This reveals that the decrease of the capacity of cycle track is closely related to the variation of the bicycle flow on the left side of cycle tracks.

(3)　In order to avoid the potential impact of platform, cyclists ride more to the right side of cycle tracks, especially at the first marking line (highest in occupancy rate).

(4)　Examining the variations in occupancy rates of the center and right of tracks (see blue dashed curves in Figure 6c–f), bicycles gradually ride back to the center at the downstream marking lines with an occupancy similar to basic segments' finally.

(5)　By comparing the occupancies of the left side of tracks in Figure 6c,d with those in Figure 6e,f, curbside stops have higher level in this data (10% vs 8.6% for 3m of track, 17.1% vs 12.7% for 4 m of track). Obviously, the bicycle traffic on the left side of tracks are most affected by the platform. More bicycles travel on the left, which means that a larger proportion of bicycles passing by are impacted (indicated clearly by operating case comparison in Figure 7), so the capacity of cycle track per meter decreases more at curbside stops.

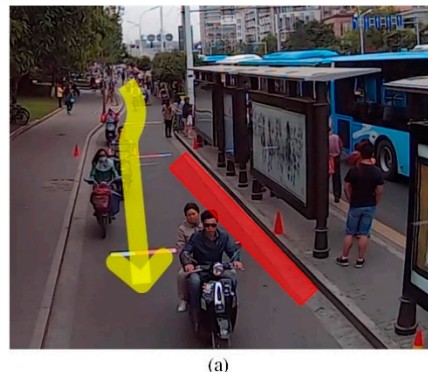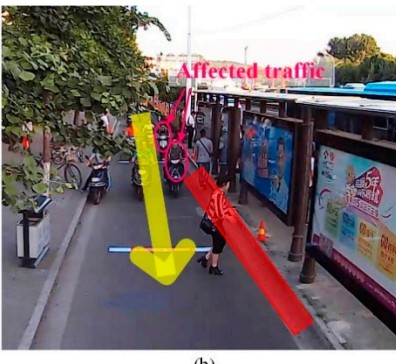

(a)                                               (b)

**Figure 7.** Comparison of real bicycle flow operating cases at (**a**) a curbside stop and (**b**) a bus bay. (yellow arrows indicate the flow trajectories, while red rectangles is the influence area of a platform).

### 4.2. Number, Location, and Density of Bus Platform Accesses

To better present the combined effects of number and locations of platform accesses, the correlation between bicycle speed and access density was analyzed in Figure 8a. It can be seen that (1) density of access results in a negative effect on capacity of track; (2) the effects of curbside stops are greater, which is indicated by the difference of line slopes.

### 4.3. Platform Length and TWPA

Through the analysis of the change of the lateral position of bicycles, it has been found that the bicycle flow is affected greatly in the upstream of a platform, and the effect gradually weakens with the riding process. That is a longer platform contributes to a smaller impact on the capacity of cycle tracks. This can be revealed by the relationships in Figure 8b. Meanwhile, the slopes of the two lines prove the curbside stops' greater impact on capacity again.

Figure 8c indicates that there is a strong positive relationship between TWPA and capacity of tracks at curbside stops, while it is weak for bus bays in the relationship.

### 4.4. TWPA to PL Ratio and TWPA to TW Ratio

As shown in Figure 8d, there are strong negative relationships between capacities of tracks at curbside stops, bus bays and TWPA to PL ratio (R = 0.975 and 0.972, respectively). The larger slope of the line fitting the data at curbside stops displays that curbside stop impact capacity of tracks more significantly.

It appears a definite positive relationship between track capacity and TWPA to TW ratio at curbside stops (R = 0.990), while the case is different at bus bays (R = 0.284). When a cyclist ride on the bypass of a curbside stop, he can only observe passengers entering and exiting through platform accesses which like windows of a house. TWPA to TW ratio represents the level of visual field a platform supplies. Thus, it is significant at curbside stops. In contrast to curbside stops, bus bays show cyclists a better visual field by the curve upstream in addition to those accesses.

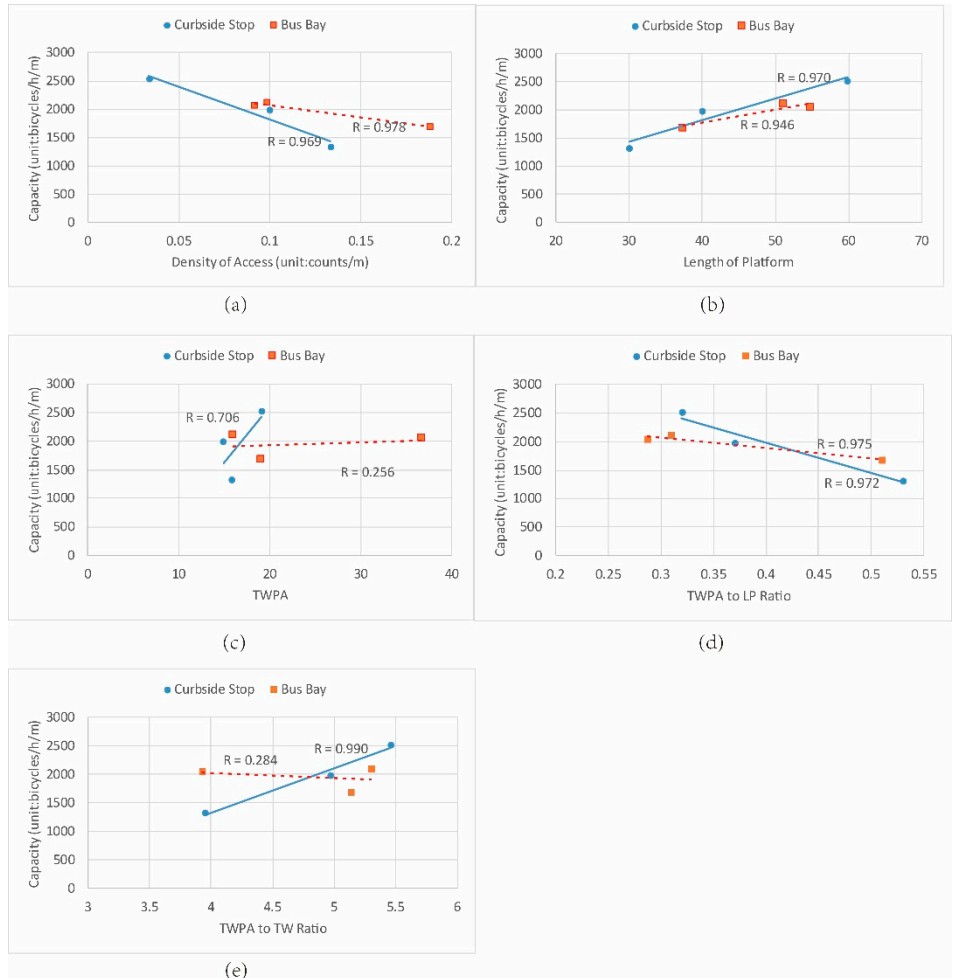

**Figure 8.** (**a**) Relationships between Capacity of Bypass and Density of Access; (**b**) Relationships between Capacity of Bypass and Length of Bus Platform; (**c**) Relationships between Capacity of Bypass and TWPA; (**d**) Relationships between Capacity of Bypass and TWPA-to-LP Ratio; (**e**) Relationship between Capacity of Bypass and TWPA-to-TW Ratio.

## 5. Suggestions for Bus Stop Configuration

By the analysis and discussions of the former two sections, we have determined the impact of bus stop type and specifications on bicycle speed and track capacity. In this section, we, first, review the basic problems of bus stop design after locating its position and, second, propose some suggestions based on the results above and setting requirements of traffic facilities, such as improving safety.

Once a stop is located, the following four basic problems facing the traffic engineers are (1) how to select the type of the stop; (2) how to determine the length of the stop; (3) how to set the number and positions of the stop accesses; and (4) how to specify the width of each access.

The research results are integrated and coordinated to propose some suggestions (indicated in Table 5) to solve the four basic problems mentioned above.

**Table 5.** Suggestions for Specifications of Bus Stop.

| Problem | Research Result | Specifying Requirement | Suggestions |
|---|---|---|---|
| how to select the type of the stop | In contrast to curbside stop, bus bay can slow the bicycles significantly, while imposing a little impact on track capacity. | Lower bicycle speed, larger capacity of track at stops, reducing the impact of bus dwelling on cars | When the land of road is enough, bus bays are highly recommended. |
| how to determine the length of the stop | A longer stop platform can reduce the influence on bicycle speed and capacity of track. | reducing the impact of stops on bicycle speed; larger capacity of track at stops | According to the number of bus lines and berths needed, a longer stop will provide a better riding circumstances for cyclists. |
| how to set the number and positions of the stop accesses | A higher density of platform access has a strong negative impact on bicycle speed and track capacity | reduce bicycle-pedestrian conflicts; larger capacity of track at stops | Accesses of bus platform should be set as few as possible and the positions need be regular. |
| how to specify the width of each access | (1) Shifting the TWPA of a stop result in reduction in bicycle speed but contribute to increase the capacity of track at curbside stops (2) A larger TWPA to PL ratio will cause reductions in bicycle speed at the two types of stops and capacity of track at bus bays. (3) A larger TWPA to TW ratio will shift the capacity of track at curbside stops | Lower bicycle speed, larger capacity of track at stops | Curbside stops need wider accesses, while keeping TWPA to PL ratio within a suitable limit; for bus bays, smaller access widths are recommended. |

(1)　How to select the type of the stop. According to our results, bus bays effectively slow bicycle flow, while they impose less impact on track capacity. Thus, bus bays better meets the specifying requirements of lower bike speed and larger capacity, which are also supported and recommended by reference [18].

(2)　How to determine the length of the stop. The analysis results shows that a longer platform impact less on both bicycle speed and capacity. So, we recommend setting a bigger stop length to suit the number of bus lines and berths needed.

(3)　How to set the number and positions of the stop accesses. Bicycle speed and track capacity are negatively impacted by dense platform accesses and fewer accesses are suggested.

(4)　How to specify the width of each access. Due to the different influence on the two stop types, wider accesses are more suitable for curbside stops, while the opposite recommendation is proposed for bus bays.

## 6. Conclusions

This paper investigated the traffic flow operation of tracks at basic sections, curbside stops, and bus bays by video recording. T-test and comparative study were used to analyze the influence of stop types on bicycle speed and capacity of track. The relationships between stop specifications and speed, capacity of track was analyzed using correlation analysis. Finally, based on the study results, some suggestions were put forward for determining stop type and selecting the parameters of a stop. The main results and conclusions are as follows:

A.　Without passengers crossing, bus bays have significant impact on bicycle speed, while it is not for curbside stops;

B.　except platform length, there are strong negative relationships between bicycle speed and density of platform access, TWPA, TWPA to PL ratio, TWPA to TW ratio;

C.　curbside stop and bus bay reduce track capacities by 32% and 13.5% on average, respectively;

D.　in contrast to bus bays, curbside stops have more significant impact on capacity of track, which also presents in the influence of the setting parameters of stops; and

E.　bus bays are recommended in selecting the type of a stop. In respect to stop specifications, a longer platform and few accesses are suggested, while wider and narrower accesses suit curbside stops and bus bays well, respectively.

Due to the limitation of investigation equipment, the length of investigated area was 50 m, which cannot cover all length ranges of bus stations. Therefore, the collected data in some sites cannot represent the bicycle flow operations completely. In addition, more bus stations need to be investigated to confirm the results of the study. The correlation analysis between bus stop parameters and bicycle speed, lane capacity was only a preliminary exploration of their relationships. Further research using other complicated methods and models like Adaptive Component Selection and Shrinkage Operator and Treed Gaussian Processes is needed in the follow-up. The two tools can well deal with the cases that discrete variables (e.g., stop type) and continuous variables (e.g., platform length) simultaneous present in one model. We will continue to improve the above defects in our other related studies.

**Author Contributions:** Z.Y. undertook the data collection. X.Y. (Xingchen Yan). provided an interpretation of the results and wrote the majority of the paper. T.W. and H.B. contributed to the paper review and editing. X.Y. (Xiaofei Ye) performed the software work of the paper. J.C. was the supervisor of the paper. All authors have read and agreed to the published version of the manuscript.

**Funding:** This research was funded by the National Key R&D Program of China (grant No. 2018YFC0704704), Natural Science Foundation of Jiangsu Province (grant No. BK20180775&BK20170932), Key Project of National Natural Science Foundation of China (grant No. 51638004), Fund for Less Developed Regions of the National Natural Science Foundation of China (grant No. 71861006), Natural Science Foundation of Guangxi Province (2019JJA160194), Guangxi Science and Technology Base and Talent Special Program (2019AC20137), Natural Science Foundation of Zhejiang Province (LY20E080011), and National Key Research and Development Program of China (2017YFE9134700).

**Acknowledgments:** The authors would like to express their sincere thanks to the anonymous reviewers for their constructive comments on an earlier version of this manuscript.

**Conflicts of Interest:** The authors declare no conflict of interest.

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
