# Peer review of "Studying the Influences of Bus Stop Type and Specifications on Bicycle Flow and Capacity for Better Bicycle Efficiency"

_information, doi:10.3390/info11080370_

Round 1

Reviewer 1 Report

A study to explore the effects and impact of type and specifications (i.e., geometric design) of bus stop on bicycle speed and cycle track capacity is carried out in this paper and it is the main subject the authors deal with. Traffic flow operations of tracks at two basic sections, three curbside stops, and three bus bays are analyzed by video recording (two cameras) in zones of Nanjing, China. Experimental results with a T-test-based comparison following a correlation analysis of the involved variables such as stop specifications, bicycles speed, and tracks capacity among others are carried out. In this sense, the authors propose that the results of this study could provide some suggestions for bus stop specifications. However, I think that the authors should make an effort to improve the paper by taking into account the following remarks:

  • Real images of the test scenarios are desirable to include for a better understanding of the analyzed characteristics in this study, thus complementing Figures 1 and 2.
  • Pictures from the cameras are desirable to include for a better understanding of the explanations of the results shown in Figures 4 and 5.
  • A better explanation related to what authors mean about further research by including “complicated” methods and models is necessary.

Author Response

Dear Professor,

We are truly grateful to your critical comments and thoughtful suggestions. Based on these comments and suggestions, we have made careful modifications on the original manuscript. All changes made to the text are highlighted in yellow. We hope the new manuscript will meet your requirements. Below you will find our point-by-point responses to your comments:

  1. Real photos of bus stops were added into Figure 1 and 2 to illustrate the traffic operating cases at bus stops and the details of field investigation.
  2. Screenshots of the typical operating cases (see Figure 7) for curbside stops and bus bays were compared to better indicate the explanations of the results shown in Figures 4 and 5 (now updated to Figures 5 and 6).
  3. We described the “complicated methods and models” from line 352 to 355.

Thank you again for your time and consideration.

Sincerely,

Authors

Reviewer 2 Report

The paper "Studying the Influences of Bus Stop Type and Specifications on Bicycle Flow and Capacity for Designing a Better Stop" introduces a methodology for desiging more efficient bus stops by taking into account observations of bicycle flows and operation of public transport. The paper is well-structured, but requires several additions to improve its representation.

First of all, all formulations should be in present tense, especially in the abstract. I would change the title, since "designing a better stop" sound trivial; here the authors could indicate who is really benefiting or what the improved parameter actually is. As optimization methods are not really applied here, I would select from efficiency, safety or comfort.

Section 2 requires and introduction, evtl. with a flow chart. Additionally, I would be beneficial to add a real frame from the video observations to show the investigation area. It is not clear how data collection and preprocessing is integral part of the methodology. Therefore a clear introduction to this section is required.

Section 5 and 6 require more content, especially in relation with table 5.

Incorporating these suggestions would highly improve the paper.

Author Response

Dear Professor,

We are truly grateful to your critical comments and thoughtful suggestions. Based on these comments and suggestions, we have made careful modifications on the original manuscript. All changes made to the text are highlighted in yellow. We hope the new manuscript will meet your requirements. Below you will find our point-by-point responses to your comments:

  1. The title was revised to “Studying the Influences of Bus Stop Type and Specifications on Bicycle Flow and Capacity for Better Bicycle Efficiency” according to your kind suggestion.
  2. To better describe Section 2, a flow chart and the corresponding indications were added. Please see line 88-98. Besides, real photos were added into Figure 2 to illustrate the details of data observation.
  3. More content in relation with table 5 have been included. Please see line 319-328 and 336-346.
  4. In the end, the formulations of the paper were checked thoroughly and corrected, especially in the abstract.

Thank you again for your time and consideration.

Sincerely,

Authors